# Three million images and morphological profiles of cells treated with matched chemical and genetic perturbations

**Srinivas Niranj Chandrasekaran** [1]**, Beth A. Cimini** [1]**, Amy Goodale** [1]**, Lisa Miller** [1]

**Maria Kost-Alimova** [1]**, Nasim Jamali** [1]**, John Doench** [1]**, Briana Fritchman** [1]**, Adam Skepner** [1]

**Michelle Melanson** [1]**, Daniel Kuhn** [2]**, Desiree Hernandez** [1]**, Jim Berstler** [1]**, Hamdah Abbasi** [1]

**David Root** [1]**, Susanne E. Swalley** [3]

**Shantanu Singh** [1]**, Anne E. Carpenter** [1]

{shsingh,anne}@broadinstitute.org
[1] Broad Institute of MIT and Harvard, 415 Main St, Cambridge, MA, 02142
[2] Merck Healthcare KGaA, Frankfurter Str. 250, 64293 Darmstadt, Germany
[3] Biogen, Inc., 125 Broadway Street, Cambridge, MA 02139.

## Abstract

We present a new, carefully designed and well-annotated dataset of images and image-based profiles of cells that have been treated with chemical compounds and genetic perturbations. Each gene that is perturbed is a known target of at least two compounds in the dataset. The dataset can thus serve as a benchmark to evaluate methods for predicting similarities between compounds and between genes and compounds, measuring the effect size of a perturbation, developing style-transfer methods to predict one experimental condition from another, and more generally, learning effective representations for measuring cellular state from microscopy images.

## 1   Introduction

Computer vision has benefitted dramatically from the revolution in deep learning. Biomedical research is an exceptionally satisfying domain on which to apply advances in machine learning, and yet deep learning applied to images in the biomedical domain has been relatively limited to medical imaging from patients, including X rays and MRI, PET, and CT scans. By comparison, deep-learning based image analysis for cell biology has generally focused on segmentation [1, 2]; whereas feature extraction and applications have lagged behind [3].

One cell biology method – image-based profiling of cell samples – is proving increasingly useful for the discovery of disease underpinnings and useful drugs [4]. In image-based profiling, human cells are cultured in samples of a few hundred cells, each sample treated with a different chemical or genetic perturbation. The resulting morphology (visual appearance) of each sample is compared by microscopy to identify meaningful differences and similarities. Among many others, applications

Submitted to the 35th Conference on Neural Information Processing Systems (NeurIPS 2021) Track on Datasets and Benchmarks. Do not distribute.

include: (a) identifying the mechanisms of a disease by comparing cells from patients with a disease to those without the disorder, (b) identifying the impact of a drug by comparing cells treated with it to untreated cells, (c) identifying gene functions or the impact of chemicals on cells by unsupervised clustering of large sets of samples to determine relationships among the perturbations tested in the experiment. Thus, image-based profiling can reveal new targets for diseases, potential therapeutics, and toxicities for particular compounds.

The vast majority of research using image-based profiling uses classical segmentation and feature extraction; deep learning methods are beginning to be explored [3] and there is much room for advancement. Historically, the lack of ground truth has been a major limiting factor in the field, as the "correct" high-dimensional profile of a given sample is unknown, and the "correct" relationships among most genes and compounds are unknown. Image-based profiling applications typically can be described as representation learning tasks; if samples are represented optimally and ideal distance metrics are applied, then biologically meaningful differences between samples will be detectable and technical artifacts will be suppressed.

To push forward advancements in this field, we assembled a consortium of ten pharmaceutical companies, two non-profit institutions, and several supporting companies, known as the JUMP-Cell Painting Consortium (Joint Undertaking in Morphological Profiling). After extensive optimization of the main assay used in image-based profiling, called the Cell Painting assay [5], this Consortium created a ground truth dataset to move methods in the field forward. We selected and curated a set of genes and compounds with (relatively) known relationships among each other, and designed an experimental layout to enable testing and comparing methods to quantify their relationships.

Here, we describe our design and creation of this dataset from a single large experiment comprising nearly three million images and over seventy five million single cells, called CPJUMP1, which contains chemical and genetic perturbation pairs that target the same genes in cells. It allows exploring a number of technical and biological parameters that might affect matching ability and testing computational strategies to match samples to each other and thus uncover valuable biological relationships.

## 2 Related datasets

We are not aware of any other Cell Painting image-based datasets that include pairs of genetic and chemical perturbations with their relationships to each other annotated, and executed in parallel so as to minimize technical variations that may confound the signal. Nevertheless, other Cell Painting datasets are public and may be useful to the community, for example as training data for self-supervised feature extraction methods. These single-perturbation-type experiments include several datasets from the Carpenter-Singh laboratory (available through the Image Data Resource [6] at `https://idr.openmicroscopy.org/search/?query=Publication%20Authors:Carpenter` and the 2018 CytoData challenge `https://github.com/cytodata/cytodata-hackathon-2018`), one from the New York Stem Cell Foundation [7] and several from Recursion, a clinical-stage biotechnology company (available at `http://rxrx.ai`).

## 3 Data acquisition

### 3.1 Compound and gene selection

Our dataset consists of images and profiles of cells that were perturbed separately by chemical and genetic perturbations, where both sets were chosen based on known relationships among them. Chemical perturbations are small molecules (i.e. chemical compounds) that modulate the function of cells while the genetic perturbations are either open reading frames (ORFs) that can overexpress genes (i.e. yield more of the gene's product in the cell) or guide RNAs that mediate CRISPR-Cas9 (clustered regularly interspaced short palindromic repeats) that can knockdown gene function (i.e. yield less of the gene's product in the cell). Most compounds are thought to inhibit the function of their target gene's product, so we expect CRISPRs to generally correlate to (mimic) the corresponding compound's profile, whereas ORFs are generally expected to anti-correlate (oppose) the corresponding small molecule's profile, and ORFs and CRISPRs targeting the same gene should generally yield opposite (anti-correlated) effects on the cells' profiles. However, we strongly note that there will be numerous exceptions given the non-linear behavior of many biological systems and a number of

distinct mechanisms by which these general principles may not hold. In fact, one aim of generating this dataset is to quantify how often the expected relationships and directionalities occur.

We derived the list of compounds from Broad's Drug Repurposing Hub dataset [8], a curated and annotated collection of FDA-approved drugs, clinical trial drugs, and pre-clinical tool compounds. The genes perturbed by genetic perturbations were chosen because they are the annotated targets of the compounds. We filtered the Repurposing Hub compounds using several criteria, of which three are important:

1. The compounds should target genes that belong to diverse gene families (Table 1). This is because the ideal methods would work well for many different biological pathways, not just a few that are well-characterized and/or easy to predict.

2. Each gene should be targeted by at least two compounds, so that gene-compound matching and compound-compound matching can both be performed using the dataset.

3. We additionally considered applying the constraint that each compound should target only a single gene. However, this criterion is difficult to achieve due to polypharmacology (Table 2), which is the property for compounds to bind and impact many different gene products in the cell; this is especially common for protein kinase inhibitors in the dataset. Instead, we only filtered out the so-called "historical compounds" listed in the Chemical Probes Portal [9], comprising compounds that are known to be quite non-selective (or not sufficiently potent) compared with other available chemical probes.

Our list of compounds and genes also includes both negative and positive controls. The negative controls for each perturbation modality are:

- Compounds: DMSO (Dimethyl sulfoxide), which is the solvent for all the compounds studied. In other words, all samples will have DMSO added at the same concentration but the negative controls have no additional compound added.
- ORFs: 15 ORFs with the weakest signature in previous image-based profiling experiments (Rohban et al., 2017).
- CRISPRs: 30 CRISPR guides that target an intergenic site (cutting controls, n = 3) or don't have a target sequence that exists in human cells (non-cutting controls, n = 27).

There are three types of compound positive controls in our list. First, we included chemical probes that are very well-studied and (unlike most compounds) are known to very selectively modulate the genes that they target [9]. Second, we included compounds that strongly correlate with the correct genetic perturbation in previous image-based profiling experiments with ORFs [10] and compounds [11]. Finally, we included a set of maximally diverse pairs of compounds with strong intra-pair and weak inter-pair correlations.

A complete description of the filtering criteria and the procedure for selecting positive and negative controls is available at `https://github.com/jump-cellpainting/JUMP-Target/`.

In the future, a commercial vendor may offer the compound set so others can test the same perturbations in other contexts for comparison.

## 3.2 Plate layout design

After applying the filters and including positive controls, we selected a total of 306 compounds and 160 genes such that they could fit into three 384-well plates, one plate per perturbation modality (compounds, ORFs and CRISPRs). Apart from a dozen or so compounds, most compounds are in singlicate. All plates included negative controls as discussed above: n=4 replicates of the 15 ORF negative controls in the ORF plate, n=2 replicates of the 30 CRISPR negative controls in the CRISPR plate, and n=64 replicates of DMSO in the compound plate. On the CRISPR plate, there are two guides per gene, each arrayed in its own well and kept separate, with no within-plate replicates. In the case of the ORF plate, for which there was only one perturbation reagent per gene, there are two replicates per plate.

We also considered the impact of edge effects, or plate-layout effects, in our design. Edge effects are the technical artifact whereby different samples will yield different behavior depending on where

Table 1: **Number of gene families with a given number of gene targets.** To maximize the diversity of genes, the genes were chosen such that most gene families (n=92) have only a single gene targeted in the final list.

| Number of gene targets (N) | Number of gene families with N gene targets in the final list |
|---|---|
| 1 | 92 |
| 2 | 16 |
| 3 | 2 |

Table 2: **Number of compounds with a given number of gene targets.** The compounds were chosen such that most compounds (n=218) in the final list are annotated as having only a single target.

| Number of gene targets (N) | Number of compounds in the final list targeting N gene targets |
|---|---|
| 1 | 218 |
| 2 | 49 |
| 3 | 23 |
| 4 | 7 |
| 5 | 4 |
| 6 | 3 |
| 7 | 1 |
| 8 | 1 |

they are located on a plate; generally this is most observed in the outer two rows and columns of the plate, and the problem persists despite efforts to mitigate it experimentally (Lundholt, Scudder and Pagliaro, 2003). While designing the plate layout, we divided the plate into outer and inner wells where the outer wells are the two rows and columns closest to the edge of the plate and the inner wells are the rest of the wells on the plate. Then we applied the following constraints in order to minimize the impact of edge effects:

1. Both of the compounds that target the same gene will either be in the inner wells or in the outer wells. They will not be split such that one of the compounds is in the inner well while the other is in the outer well.

2. The gene target of outer well compounds will be in the outer wells of the genetic perturbation plate.

3. All the positive control compounds are in the inner wells.

If preferable, an analysis can be constrained to the inner wells only, to ensure that edge effects have minimal influence on the results.

### 3.3 Experimental conditions

We acquired our data under the following experimental conditions:

1. Four replicate plates of compounds and CRISPRs and two replicate plates of ORFs (which, as mentioned, contain two replicates within each plate) at two time points and two cell lines each. The short and long time points were different for each perturbation type: compounds (24-hour, 48-hour), ORFs (48-hour, 96-hour) and CRISPRs (96-hour, 144-hour). The two cell lines were U2OS and A549.

2. One plate of the A549 96-hour ORF plate where the cells have been additionally treated with Blasticidin (a drug that kills cells that have not been properly infected with the genetic reagent).

3. Two replicate plates of the A549 144-hour CRISPR plate where the cells have been additionally treated with Puromycin (a drug that kills cells that have not been properly infected with the genetic reagent).

4. Two replicate plates of the A549 48-hour compound plate with 20% higher cell seeding density than the baseline.

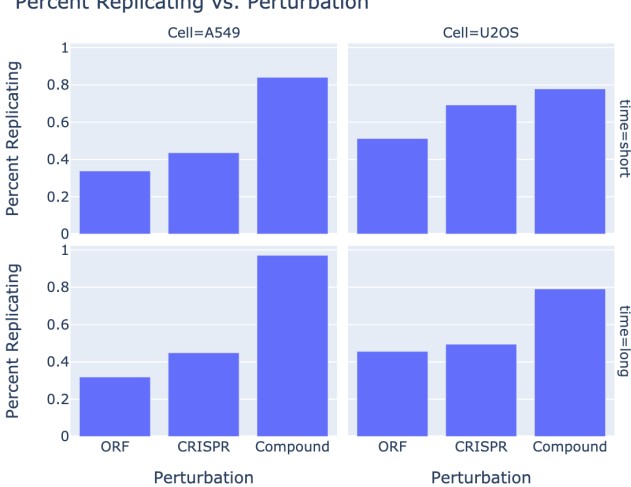

Figure 1: ***Percent Replicating* vs. perturbation modality.** Compounds have a stronger within-replicate correlation compared to ORFs and CRISPRs.

5. Two replicate plates of the A549 48-hour compound plate with 20% lower cell seeding density than the baseline.

6. Four replicate plates of the A549 24-hour compound plate were imaged six additional times to test photobleaching from repeated imaging.

7. Two replicates of the ORF plates in U2OS and A549 at 96-hour and 144-hour were imaged four additional times, once on each of days 1, 4, 14, 28 after the first imaging, to test the stability of samples over time.

# 4 Potential uses

The CPJUMP1 dataset was designed to test several experimental conditions to determine which yield the highest signals and best matching ability. We will establish best practices for the laboratory work based on our analysis of these results, not further detailed here (Cimini et al., in preparation). Here we focus on the applications that are most of interest to a machine learning audience.

## 4.1 Benchmarking perturbation-detection methods

Detecting which samples are measurably different from negative controls is one task that often precedes other useful applications, and is equivalent to measuring the effect size. For example, a set might be filtered by this criterion before embarking on subsequent laboratory experiments, or prior to training a model, or other analysis that could be confounded by noisy signals. It can also be useful for determining what experimental protocol or computational analysis pipeline to use among several alternatives. It should be noted that even given perfect computational methods for feature extraction, batch correction, and profile comparison, not all samples will be detectably different from negative controls for several biological reasons. For example, a drug or genetic perturbation may only impact cell morphology in a particular cell type, under particular environmental conditions, at a particular time, or if particular stains were used, conditions which may not have been met in the experiment.

To detect the number of samples with a measurably distinct phenotype, we estimated *Percent Replicating* (Figure 1), which is the proportion of samples that are distinct from the null distribution built from samples that are non-replicates. A sample is considered to have a detectable signature if the median of the correlation between the replicates of the sample is greater than the 95th percentile of the null distribution. In other words, *Percent Replicating* is the True Positive Rate if the False Positive Rate is set to 5%, for a binary classification problem where replicates make up the positive class and non-replicates make up the negative class.

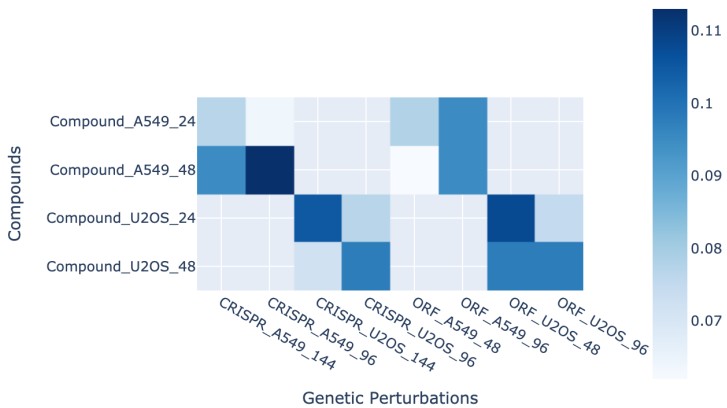

Figure 2: *Percent Matching* **between compounds and genetic perturbations.** Axis labels include the cell lines (A549 or U2OS) and the timepoints (48 hour, 96 hour, and 144 hour).

## 4.2 Benchmarking gene-compound matching methods

This dataset presents a unique opportunity to match profiles of perturbations across modalities (chemical versus genetic), because genes in this dataset that are targeted by two types of genetic perturbations (ORF and CRISPR) are also targeted by two compounds. To establish a baseline approach to match profiles across modalities, we computed the Pearson correlation between all chemical and genetic perturbation pairs. We then evaluated the performance of our approach by estimating *Percent Matching* (Figure 2), which is the proportion of "true" connections (chemical-genetic perturbation pairs that target the same gene) that are distinct from a null distribution built from "false" connections (chemical-genetic perturbation pairs that are not known to target the same gene). A true connection is considered to be correctly detected if its correlation is greater than the 95th percentile of the null distribution. In other words, *Percent Matching* is the True Positive Rate if the False Positive Rate is set to 5%, for a binary classification problem where the true connections make up the positive class and the false connections make up the negative class.

The baseline results show that there is a signal in this dataset for matching chemical and genetic perturbations that target the same gene ( 7-11%, against a false positive rate of 5%), but there is much room for improvement. It should be strongly noted, though, that significant time and resources can be required to identify the target of a compound, and similarly to identify compounds that target a particular gene. Therefore, these low rates may already be highly meaningful, and improvements in image representations and measuring similarities could have a major impact on the pharmaceutical industry.

Given this dataset also has pairs of compounds targeting the same gene, it can also be used to test compound-compound matching.

## 4.3 Benchmarking style transfer methods

The design of CPJUMP1 included multiple cell types, timepoints, modalities (compound, ORF, and CRISPR), imaging conditions, and selection conditions. This allows the unusual opportunity to attempt prediction of one experimental condition from another. There are many potential combinations here, so we do not provide a baseline but simply point out this possibility to the interested researcher.

## 5  Code and Data availability

Cell images, morphological profiles, image analysis pipelines, profile generation pipelines, plate maps and plate and compound metadata are available online at `https://broad.io/neurips-cpjump1`.

The data used to generate the figures are available online. Figure 1: `https://github.com/jump-cellpainting/neurips-cpjump1/tree/main/analysis#percent-replicating` and Figure 2: `https://github.com/jump-cellpainting/neurips-cpjump1/tree/main/analysis#percent-matching-across-modalities`.

# 6 Methods

## 6.1 Sample preparation and image acquisition

The Cell Painting assay involves staining eight components of cells with six fluorescent dyes: nucleus (Hoechst), nucleoli and cytoplasmic RNA (SYTO 14), endoplasmic reticulum (concanavalin A), Golgi and plasma membrane (wheat germ agglutinin; WGA), mitochondria (MitoTracker), and the actin cytoskeleton (phalloidin). We optimized the Cell Painting assay described in (Bray et al., 2016) by changing the concentrations of Hoechst, phalloidin, concanavalin A and SYTO14 and combining dye addition and dye permeabilization steps. These changes will be described in more detail in (Cimini et al., in preparation) and are currently publicly available at `https://github.com/carpenterlab/2016_bray_natprot/wiki#updates-to-the-cell-painting-protocol`. The images were acquired across five fluorescent channels using a Perkin Elmer Opera Phenix HCI microscope at 20x magnification.

## 6.2 Image processing

We used the CellProfiler [12] bioimage analysis software to process the images. We corrected for variations in background intensity, and then segmented cells, distinguishing between nuclei and cytoplasm. Then, across the various channels captured, we measure various features of cells across several categories including fluorescence intensity, texture, granularity, density, location (see `http://cellprofiler-manual.s3.amazonaws.com/CellProfiler-3.0.0/index.html` for more details). Following the image analysis pipeline, we obtain more than 75 million cells and 5792 feature measurements.

## 6.3 Image-based profiling

We used *cytominer* (`https://cytomining.github.io/profiling-handbook/`) and *pycytominer* workflows (`https://github.com/jump-cellpainting/profiling-recipe`) to process the single cell features. We aggregated the single cell profiles by computing the mean. We then normalized the averaged profiles by subtracting the median and dividing by the median absolute deviation (m.a.d.) of each feature. This was done in two ways: using the median and m.a.d. of (i) the negative control wells on the plate (used in the analysis shown here), and (ii) all the wells on the plate. Finally, we filtered out redundant features as well as features with low variance. All the steps in the profiling workflow were performed for each individual plate separately.

## Acknowledgments and Disclosure of Funding

The authors appreciate the more than 100 scientists who have contributed to the organization and scientific direction of the JUMP Cell Painting Consortium. We thank Max Macaluso (operations) and Tanaz Abid (technical) at the Broad Institute for their assistance as well.

The authors gratefully acknowledge a grant from the Massachusetts Life Sciences Center Bits to Bytes Capital Call program for funding the data production. We appreciate funding to support data analysis and interpretation from members of the JUMP Cell Painting Consortium and from the National Institutes of Health (NIH MIRA R35 GM122547 to AEC). The authors also gratefully acknowledge the use of the PerkinElmer Opera Phenix High-Content/High-Throughput imaging system at the Broad Institute, funded by the S10 Grant NIH OD-026839-01.

AEC has optional ownership interest in Recursion, a public biotechnology company using image-based profiling for drug discovery. SES is an employee of Dewpoint Therapeutics. Daniel Kuhn is an employee of Merck Healthcare KGaA, Darmstadt, Germany.

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

# A    Appendix

The landing page of the GitHub repository for this dataset has all the relevant additional information:
`https://broad.io/neurips-cpjump1`.

We have released the data with a CC0 licence and the code with a BSD 3-Clause license.

We have chosen GitHub as the hosting platform, and use GitLFS to store large files.

