# OpenReview forum: "Three million images and morphological profiles of cells treated with matched chemical and genetic perturbations"
_NeurIPS.cc/2021/Track/Datasets_and_Benchmarks/Round1 — Submitted to NeurIPS 2021 Datasets and Benchmarks Track (Round 1)_

### Official Review · Reviewer_v8RB · 2021-07-04
**This work mainly proposes a new image dataset, but it is mainly analyzed from the genetic and chemical perspective, so I think this paper needs to be improved.**

**Rating:** 4
**Confidence:** 2
**Clarity:** The organization of this papar may be…

**Strengths:**

1. The dataset can serve as a benchmark to evaluate methods for predicting similarities between compounds and between genes and compounds and measuring the effect size of a perturbation.
2. It can also be applied to the field of machine learning, such as perturbation-detection methods, gene-compound matching methods and style transfer methods.

**Weaknesses:**

The dataset is mainly analyzed from the aspects of chemistry and biology, and only introduces the potential use of machine learning in general, but does not introduce the relevant experimental results and the contribution in detail.

**Additional Feedback:**

The images in this dataset may be divided into a training set, a validation set and a test set in reasonable way for the applications of machine learning.

**Correctness:**

The dataset obtained under some experimental conditions are introduced. It includes different cell lines (A549 or U2OS) and different timepoints (48 hour, 96 hour, and 144 hour). It may be constructed in a sound way.

**Documentation:**

For this dataset, this work include the document and intended use, as well as the URL to access the dataset.

**Ethics:**

There are no ethical concerns for the submission.

**Relation To Prior Work:**

No, this work does not compare the proposed dataset with previous contributions.

**Summary And Contributions:**

This work present a new and well-annotated dataset called CPJUMP1 which can serve as a benchmark to evaluate methods for predicting similarities between genes and compounds. It allows the exploration of many biological parameters that may affect the matching ability and testing computational strategies to match samples to each other and thus uncover valuable biological relationships.

---

### Official Review · Reviewer_Ju3L · 2021-07-04
**Cell image/profile dataset with potential but insufficient demonstration**

**Rating:** 4
**Confidence:** 4

**Strengths:**

The CPJUMP1 dataset presented in this paper is made possible by a remarkable consortium of ten pharmaceutical companies, two non-profit institutions, and several supporting companies. The dataset contains millions of images and morphological profiles, and it is fully annotated with the relationships among the genes and chemical compounds provided as ground truth. Therefore it has the potential to be used by the ML and biomedical research community to uncover valuable biological relationships.

**Weaknesses:**

First of all, the manuscript spends too much effort describing the technical details about how this dataset is acquired but too little information regarding how this dataset is used and benchmarked and how it can be useful to the ML and biomedical research community. It would be much better if the authors could place most of the technical details in the supplementary materials but spend more effort benchmarking their dataset and convincing the reader that the dataset is impactful. Unfortunately, out of millions of images and profiles, the authors only present two simple benchmark figures (percent replicating and percent matching) with very limited information and insights. If the authors themselves cannot demonstrate how to fully explore the abundant information in the dataset, it might be even harder for the reader to find that information useful.

Second, as disclosed by the authors in the Related Datasets section, there are already more than 10 similar publicly available datasets that contain images and morphological profiles with annotated genetic and chemical perturbation relationships. Therefore, besides having more images/profiles and annotating more perturbation pairs, I do not see a significant improvement in this work over the existing ones. Moreover, while those existing datasets are publicly available and free, the CPJUMP1 dataset is currently hosted on AWS and repairs the user to pay for downloading the data. The dataset is also not user-friendly as the tutorial is not well illustrated.

Third, the description of the data acquisition process is vague and confusing. For example, the authors mention that they selected a total of 306 compounds and 160 genes such that they could fit into three 384-well plates, but they did not provide details regarding how the selection was made. Also, the authors consider the impact of edge effects, but they did not describe these effects in detail and investigate how these effects would affect the results. In addition, the authors use different perturbations on different cell lines with different numbers of replicates, but they did not provide the reason for doing so. These combinations seem to be random but not systematic. Thus, there are so many variables in the dataset and it is difficult to consider a single variable at a time. Finally, the description of image processing and image-based profiling is not clear.

Besides the main problems above, there also minor issues.

- As a manuscript describing an image/profile dataset, there is not a single figure showing the raw data, i.e., the cellular images and their morphological profiles.

- There are many places where the authors add citations with texts instead of numbers in square brackets as they are supposed to be, e.g., lines 163, 222, 224.

- The authors fail to describe any limitations of their dataset.

- All images in the dataset are acquired across five fluorescent channels using a Perkin Elmer Opera Phenix HCI microscope at 20x magnification. This can be problematic as there could be unnoticed systematic artifacts with this microscope. It would be wise to add images acquired with different microscopes to the dataset.


**Additional Feedback:**

The authors are recommended to provide more technical details of their dataset in the supplementary materials, not in the main text. Then, in the main text, the authors are recommended to provide more benchmark experiments that really explore the potential of this dataset. The authors already point out several potential usages of the dataset, such as quantifying how often the expected relationships and directionalities occur and predicting one experimental condition from another. Instead of simply pointing out these possibilities, the authors are recommended to demonstrate these possibilities themselves to really convince the reader that the dataset is different from previous ones and it can be useful.

**Clarity:**

The paper is most clearly written, but the lack of details and demonstrations impairs its clarity.

**Correctness:**

The claims made by the authors seem to be correct, but they fail to provide enough details and proof to support their claims.

**Documentation:**

- The detail on data collection and organization is insufficient. The authors are recommended to describe them with more details in either the supplementary materials of the GitHub repository.

- The data are currently not free to download. They are hosted on AWS and require the user to pay for the download.

- The documentation is not user-friendly. The tutorial of using the data is unclear.

- The benchmarks experiments are insufficient.

**Ethics:**

I do not see any ethical concerns in this dataset.

**Relation To Prior Work:**

The authors mention the previous datasets that have similar perturbations. They claim that the previous datasets only have single-perturbation-type experiments while their dataset has pairs of perturbation. However, due to the lack of details and demonstrations, I find it hard to conclude that this work is really different from the previous datasets.

**Summary And Contributions:**

The authors present a large cellular image and morphological profile dataset called CPJUMP1 which contains chemical and genetic perturbation pairs that target the same genes in cells. This dataset is created based on the Cell Painting technique, and it is the first such dataset that includes annotated information about the pairs of genetic and chemical perturbations and their relationships. Whereas the dataset has the potential to be useful for the ML and biomedical research community, the manuscript and its GitHub repositories did not provide enough details regarding how this dataset can be easily accessed and used by other researchers.

---

### Official Review · Reviewer_B1c3 · 2021-07-04
**A large scale dataset of cell profiles, with inadequate benchmarks and missing details.**

**Rating:** 4
**Confidence:** 3

**Strengths:**

The paper presents a novel dataset of morphological cell profiles, CPJUMP1, obtained during a large experiment campaign where both chemical and genetic perturbations are considered under different settings. This has not been seen before, and is therefore very novel. The large dataset could lead to researcher from the machine learning community to consider working on profiling of cell samples. An URL is provided for easy data and code access, and he license and hosting platform are described in the supplementary materials.

**Weaknesses:**

Neither good nor bad ethical or societal implications are discussed. It would be worth dedicating some lines in the paper to this, as it appears that this dataset could bootstrap machine learning within is this field, which will most likely have some good societal implications.

**Additional Feedback:**

Given that the paper proposes an image dataset, it stands out that not a single image of the dataset is present in the paper or supplementary materials. This makes it hard for the reader to understand the type of data is considered.
In order to improve the paper, the remaining two pages of the nine allowed should be utilized for adding extra details. The tasks should contain more benchmarks from representative fields of each of the proposed tasks. Furthermore, the metrics should be justified, and if there are any hyperparameters in the benchmarking methods these should be searched


The authors have decided not to update the current manuscript, and instead opt to submit again for the second submission deadline of the dataset track. Here they promise to clarify the aspects mentioned by the reviewers. Therefore, my rating is unchanged as of now.

**Clarity:**

The paper is well written. However, the paper does not contain a full checklist, with only parts filled out.

**Correctness:**

The fundamental experiment design is very well described, in detail, and covers many different scenarios, and documents choices for both compound and gene selection as well as plate layout. However, the dataset will target partly the medical imaging community as well as the computer vision community, it would make sense to explicitly state:
1) How the data is distributed into different dataset splits.
2) How many images there are of each condition instead of the amount of plates.
3) Whether the data is labeled on an image level, with e.g. bounding box or segmentation masks, or are the annotations only for the genetic and chemical compound relationships.

Three different use cases for the data are proposed:
1) An unsupervised task for detecting whether the data differs from a negative control after been subjected to a perturbation.
2) A gene-compound matching task is proposed, where the cell profiles of genes targeted by different genetic and chemical perturbations are to be matched.
3) A style transfer task based on the different modalities and settings of the data

For task 1 and 2 simple benchmarks are proposed. For task 1 a benchmark results are obtained median of replicates compared to the null distribution, and for task 2 a benchmark results are obtained using the Pearson correlation and the null distribution based on examples of chemical and genetic perturbations which do not target the same genes.

There are several problems with the benchmarks. The methods are not clearly described, as the entire image processing pipeline has been skipped. It is mentioned in section 6.2 that CellProfiler is used, but it is not clear whether the benchmarks are performed on a per-cell or per-plate basis and what exact methods are used. Task 1 is basically an anomaly detection problem, and therefore it would be relevant to compare with state of the art anomaly detection methods. Task 2 is similar to the re-identification task and it would therefore be relevant to compare with methods from the re-identification and metric learning domains. Furthermore, there are no baseline for task 3, meaning the task is actually not considered and seems irrelevant at the moment.
It is also not clear why the metrics for task 1 and 2 were decided upon instead of standard metrics such as F1 score, Matthews Correlation Coefficient, Mean Average Precision etc. For task 3 no evaluation metric was described.


**Documentation:**

The dataset is well documented, and available. Code has been provided on a GitHub page with descriptions of how to download the data and how to run analysis script, together with descriptions the used compute resources. It is not clear how to reproduce the benchmark results.

**Ethics:**

There are no ethical concerns for the submission.

**Relation To Prior Work:**

It is clear how the dataset is positioned compared to previous datasets, and how the cell profiling use cases is different from previous biomedical use cases.

**Summary And Contributions:**

The paper proposes a very large novel image-based dataset for investigating the effect on cells when treated with chemical and genetic perturbation, under a large variety of experiment settings. A large amount of perturbation variations is obtained and compared, with both negative and positive controls, and each gene is target buy at least two different compounds to enable gene-compound and compound-compound matching.

---

### Author Response · Authors · 2021-07-14
**Response to reviewers**

We thank the reviewers for their detailed comments, which we believe will significantly improve the quality of the paper.

In our Round 2 submission, we will address all of the following

- **Discuss societal implications** – methods that improve the efficiency of the presented applications can dramatically accelerate drug discovery and therefore improve human health and reduce drug development costs. We will explicitly discuss this and include statistics about the economic burden of finding new medicines. As Reviewer 1 points out, it will also attract attention to the use of computer vision and machine learning in service of discovering new medicines, which is a relatively underpopulated field.

- **Provide more details about the dataset** – the reviewers had several questions about the details regarding experimental design, data acquisition, data processing, and metadata; we will address each and then have the paper the GitHub README tutorial reviewed by machine learning colleagues unfamiliar with our domain for clarity. We will also complete the checklist, fix reference formats, and provide example images from the assay, showing a matched set: cells treated with a drug and with genetic perturbations that all target the same pathway, alongside untreated cells for comparison.

- **Contextualize the importance of this dataset relative to existing data** – we will explain why this dataset is so exciting and useful: primarily, it is the first experiment to simultaneously assess three completely different ways to perturb cell pathways: chemical compounds (drugs), gene overexpression (ORFs), and gene knockdown (CRISPR). It is the first dataset systematically built to study the relationship between matched pairs of drugs and gene perturbations; prior datasets may have annotations for a subset of genes and drugs, but these rare samples are spread across many experimental batches and plates; here we have concentrated all relevant pairs into a single well-controlled experiment.

- **Clarify data access** – we had set up a “requester pays” access at the time of submission but had failed to clarify that this will be made available with AWS Open Data soon; we will clarify this and add more details.

- **Address limitations** – we will further clarify the limitations of such experiments and cite review papers on the topic that discuss this in more detail. Namely, an ideal experiment would include more samples (this is limited by cost), sample preparation and image acquisition at multiple separate laboratories and different microscopes and other instrumentation (limited by cost) and perfect annotations of which drugs impact which cell pathways (limited by existing, imperfect human knowledge).

- **Contextualize the relevance of the benchmarks to drug discovery** – we will provide more details on the two primary use cases (perturbation detection and gene-compound matching) of the datasets. We will also better motivate the importance of these tasks on drug discovery; we realize this is definitely not obvious to those outside the domain. Further, we will address the relevance of anomaly detection and reidentification to the problems at hand.

- **Provide alternative metrics** – we will additionally report information retrieval metrics alongside the metrics already included. Our choice of metrics was based on decisions made downstream; typically a fraction of the perturbations or connections among perturbations are followed up on, and these metrics give a direct readout of what this fraction is.

---

### Decision · Program_Chairs · 2021-07-26

**Decision:**

Reject

**Comment:**

The reviews feel that the dataset is interesting, but the contributions are not sufficient for acceptance. (1) The methods, including data acquisition and benchmark development, are not clearly described. (2) The authors did not prove the contribution of this work over existing datasets.